# A Theoretical Model with the Effect of Cracks in the Local Spalling of Full Ceramic Ball Bearings

**Huaitao Shi [1,2], Zimeng Liu [1], Xiaotian Bai [1,2,]\*, Yupeng Li [2] and Yuhou Wu [2]**

[1]   School of Mechanical Engineering, Shenyang Jianzhu University, Shenyang 110168, China;
      sht@sjzu.edu.cn (H.S.); liuzm@stu.sjzu.edu.cn (Z.L.)
[2]   Joint International Research Laboratory of Modern Construction Engineering Equipment and Technology,
      Shenyang, Liaoning 110168, China; lyp@sjzu.edu.cn (Y.L.); wuyh@sjzu.edu.cn (Y.W.)
\*   Correspondence: baixt@sjzu.edu.cn

**Abstract:** For full ceramic ball bearings, cracks occur frequently in the spalling on the rings, which leads to impacts on the bearing dynamic characteristics. In this paper, the spalling is set on the outer ring, and the dynamic model considering the effect of crack is proposed. The crack is considered to be related to the strain energy, and the effect on the stiffness of the outer ring is also analyzed. Results show that the appearance of cracks leads to the reduction of the full ceramic bearing stiffness, and the vibration amplitude of bearing increases. The effect of a crack depends on its size, and the vibration of the bearing with cracks of different widths and depths vary greatly. This study provides theoretical basis for the calculation of full ceramic bearing and is of great significance for the state monitoring and fault diagnosis.

**Keywords:** full ceramic ball bearing; outer rings; crack; spalling; nonlinear dynamic model

## 1. Introduction

Compared with steel bearings, full ceramic bearings have been widely studied because of their lighter weight and higher hardness. The performances of full ceramic bearings and hybrid ceramic bearings have been compared with those of steel bearings and have showed advantages in speed [1] and service life [2–5]. Full ceramic bearings have been widely used in high-speed lathes, aero-engines, chemical machinery, and other fields. In the special environment of high-speed heavy-duty, low noise, and high precision, full ceramic bearings have gradually taken the main positions. The main failure modes of full ceramic bearings are spalling and cracking. Compared with steel, the ceramic material is higher in brittleness and lower in fracture toughness, so spalling occurs more frequently on the contact surface under high Hertz pressure, which leads to small cracks around the surface defects [6–10].

The study of the effect of the local spalling cracks on the vibration response of full ceramic bearings is valuable for the health monitoring of full ceramic bearings during operation. The spalling defect is more complicated on the rings of full ceramic bearings than on steel bearings as the failure modes of ceramic material are different from the steel modes. [11] Cracks usually appear in spalling areas, and as a result, the model with the effect of cracks in spalling is more accurate. At the same time, considering that the ceramic material is more sensitive to the local damage, the analysis on the vibration signals of the outer ring can be of great help for the fault detection on the outer ring.

The majority of these models [12–14] are designed to model bearings with line spalling defects that occur at the early stages of bearing failure. The extended spalling defects that occur at later stages due to the successive rolling element that passes over the defect have received very little attention [15]. Most models of bearing outer ring spalling adopt the method of establishing the geometric model of flaw and simulating the deformation of flaw to calculate the time-varying rigidity of bearing outer

ring fault [16–19]. Because the hardness and the fracture toughness of ceramic materials are very different from those of steel materials, the contact deformation of ceramic bearing outer ring cannot be accurately simulated by establishing the geometric model of spalling defects. Previous research [20–22] has deeply discussed the cracks in ceramic ball bearings, and it considered that the cracks were caused by the contact stress between the rolling elements and the bearing rings during the operation of the equipment. However, the bending stress exerted by the rolling elements on the outer ring during the process of passing through the spalling position of the outer ring was not taken into account in the study, and the bending stress may also be the reason for the initiation and propagation of the cracks. Researchers [23–28] have conducted deep analyses on the crack formation mechanism of the cracks under contact stress. On the basis of the principle of fracture mechanics, the influence of cracks caused by bending moment on stiffness can be analyzed by calculating the strain energy caused by cracks [29–31]. According to the failure principles of ceramic materials, when the spalling occurs on the ring of the full ceramic ball bearing, cracks also appear in the spalling area. The generation and propagation of cracks are significant to the dynamic characteristics of the full ceramic bearing. However, studies on the effect of the cracks in spalling are quite rare so far, to the author's knowledge.

In this paper, an improved nonlinear dynamic model with the effect of cracks in spalling is presented, which can be used to analyze the vibration responses of bearings with cracks and spalling on the outer ring. The modified nonlinear model takes into account both the modified rigidity of bearing outer ring flaking fault, which is proposed by Eppsik [32], and the load modulation of local fault pulse sequence proposed by Mcfadden and Toozhy [33]. It has the advantage of avoiding the estimation of the contact deformation at the flaw and reducing the influence of different mechanical properties of silicon nitride ceramics and steel. On the premise of the spalling failure of ceramic bearing outer ring, the effect of cracks of different sizes at the spalling position on the stiffness of bearing outer ring was calculated; in calculating the stiffness of the outer ring, the strain energy change caused by the crack is taken into account. In this way, the influence of the spalling fault on the contact rigidity can be studied more comprehensively under both contact and non-contact loads. The effect of cracks of different sizes is also discussed, and parametric studies on the depth and width of the cracks are conducted. The impacts of crack sizes on the dynamic characteristics are analyzed quantitatively, and the conclusions are drawn at last.

## 2. Nonlinear Multi-Body Dynamic Modelling of a Defective Bearing

### 2.1. Kinematics of the Rolling Elements

In some applications like in backup bearings, the inner ring is fixed, therefore it might be stated that during the operation of most equipment, the outer ring is usually fixed on the bearing seat and rotates along with the rotating shaft. It is not ruled out that there are individual inner rings fixed and the outer ring fixed on the rotor rotating along with the shaft. For the sake of simplicity, assuming that there is no sliding friction between the rolling element and the inner and outer raceways, the position angle of the *j*th rolling element can be expressed as [34]

$$\theta_j = \frac{2\pi(j-1)}{Z} + \theta_1 + \omega_c t \tag{1}$$

where $Z$ is the number of rolling elements, $\theta_1$ is the initial angular position of the first rolling element, and the nominal rotational speed of the cage is given by

$$\omega_c = \frac{\omega_s}{2}\left(1 - \frac{d}{D} \cdot \cos\alpha\right) \tag{2}$$

where $\omega_s$ is the shaft run speed, $d$ is the diameter of a rolling element, $D$ is the pitch diameter, and $\alpha$ is the contact angle.

The impact pulse sequence is generated when the bearing stripped part contact is periodic, and the period depends on the stripped part. According to the kinematics analysis of geometric conditions, the failure passing frequency of the bearing outer ring can be obtained by:

$$f_{\text{outer}} = \frac{f_a \cdot Z}{2}\left[1 - \left(\frac{d}{D} \cdot \cos\alpha\right)^2\right] \cdot \frac{d}{D} \tag{3}$$

where $f_a$ is the frequency of rotation of shaft.

When a vertical downward radial force is applied to the shaft while ignoring the effect of the force on the upper half of the rolling element of the bearing, as shown in Figure 1, the load distribution received by the outer ring of the ball bearing under the action of the radial force can be expressed as:

$$Q_\varphi = \begin{cases} Q_{\max}\left[1 - \frac{1}{2k_L}(1 - \cos\varphi)\right]^{\frac{3}{2}} & \varphi \in \varphi_{\text{load}} \\ 0 & \text{others} \end{cases} \tag{4}$$

where $Q_{\max}$ is the maximum load distribution density, $\varphi$ is the arbitrary position angle in load area, and $k_L$ is the load distribution coefficient, which can be expressed as:

$$k_L = \frac{1}{2}\left(1 - \frac{c}{2\delta_{\max}}\right) \tag{5}$$

where $c$ is the ball bearing clearance and $\delta_{\max}$ is the maximum radial offset of bearings.

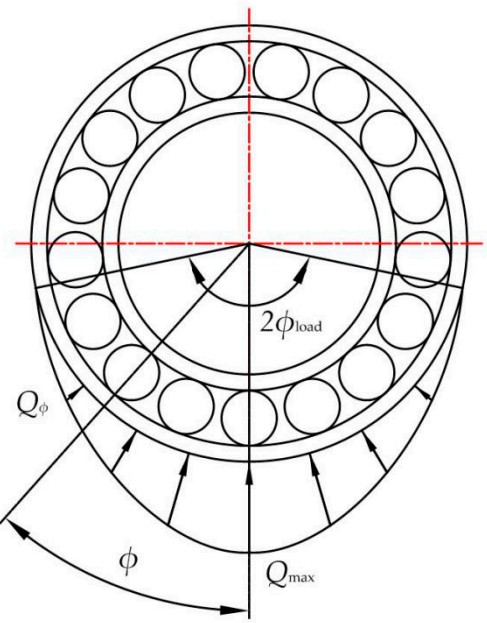

**Figure 1.** Load distribution diagram of outer ring during bearing operation.

### 2.2. Modified Time-Varying Stiffness of Local Spalling Faults

Unlike the steel outer ring, the surface of the ceramic outer ring will not undergo obvious local deformation when full ceramic bearings produce local spalling failure. Therefore, it is not suitable for the traditional Hertz contact theory in the area where the rolling element contacts spalling, and the contact stiffness value in this area needs to be revised. As can be seen in Figure 2, when the rolling element operates within the spalling fault range (the rolling element enters the edge and does not come into contact with another spalling edge), the position of the rolling element is $\beta_{\text{en}} < \beta < \beta_f$. The new

contact stiffness $K_{o1}$ is obtained by modifying the contact stiffness $K_o$ between the outer rings, and the outer rings can be obtained by:

$$K_{o1} = K_o\left[1 - R\left(1 + \left(\frac{\beta_{en} - \beta}{\beta_{en}}\right)\right)^S\right] \tag{6}$$

where $K_o$ is the contact stiffness between rolling element and outer ring—its calculation formula can be found in reference [35]—$\beta_f$ is the angular position of spalling fault center, $R$ is the reduction factor, the reduced ratio of load–deformation contacts in the range $\beta_{en} < \beta < \beta_f$, of which the parameter is usually expressed as a percentage, and $R > 50\%$ is taken; $S$ is the shape factor and can be any value.

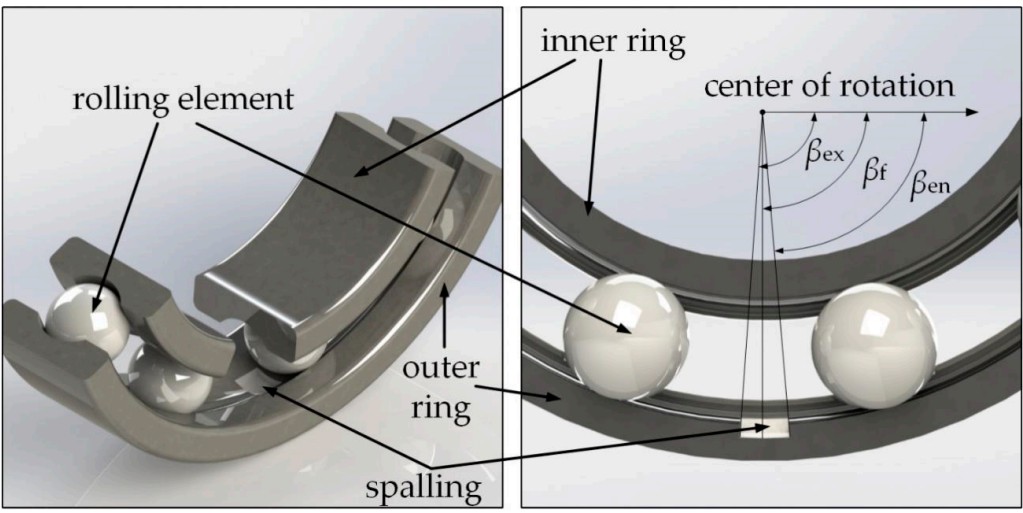

**Figure 2.** Angular diagram of the position of the rolling elements passing through the bearing outer ring spalling.

As the rolling element operates within the range of spalling faults, the corresponding $K_{o1}$ values differ at different locations. When the rolling element is in a spalling fault (contacting both edges of the spalling), the position of the rolling element is $\beta = \beta_f$, and the modified contact stiffness between the rolling element and the outer ring of the ceramic bearing can be expressed as:

$$K_{o1} = K_o(1 - R). \tag{7}$$

As that rolling element leaves the spalling area, the rolling element is restored to contact only one side of the spalling fault position, where the position of the rolling element is $\beta_f < \beta < \beta_{ex}$, and the modified contact stiffness can be expressed as:

$$K_{o1} = K_o\left[1 - R\left(1 + \left(\frac{\beta - \beta_{ex}}{\beta_{ex}}\right)\right)^S\right]. \tag{8}$$

To sum up, by modifying the value $K_{o1}$ in different states, the time-varying stiffness of the outer ring can be modified to make the load–deformation relationship more consistent with the actual situation of the contact between the rolling element and the ring, and also make the subsequent modeling more authentic.

According to the literature, the total contact stiffness $K$ of the bearings can be obtained, and the calculation formula is expressed as follows:

$$K = \left[\frac{1}{(1/K_i)^{2/3} + (1/K_{o1})^{2/3}}\right]^{3/2} \tag{9}$$

where $K_i$ is the contact stiffness between the rolling element and the inner ring; the calculation formula of $K_i$ can be referred to reference [35].

According to the Hertz contact theory, the contact deformation between the rolling element and the outer rings can be obtained when the rolling element passes through the stripping fault region:

$$\delta_{os} = \left(\frac{Q_\varphi}{K_{o1}}\right)^{-3/2}. \tag{10}$$

### 2.3. Bearing Modulation of Local Fault Pulse Sequence

Considering that the ceramic material is different from common steel material, the ceramic material has high brittleness, the contact deformation between bearing outer ring and rolling element is different from that of steel bearing in the process of establishing full ceramic bearing outer ring spalling fault model, so the impact load on the spalling fault position is studied in this paper. The results show that ceramic material has high brittleness and the contact deformation between bearing outer ring and rolling element is different from that of steel bearing when spalling fault occurs. In order to approach the actual working condition, it is assumed that the impact force produced by the contact between the bearing rolling element and the fault part is a rectangular pulse under the unit load. So that those shock sequences accompany the bearing motion can be represented by infinite rectangular functions of equal amplitude within a period, as shown in Figure 3, the pulse signal may be regarded as a trapezoidal signal when the pulse interval time is short.

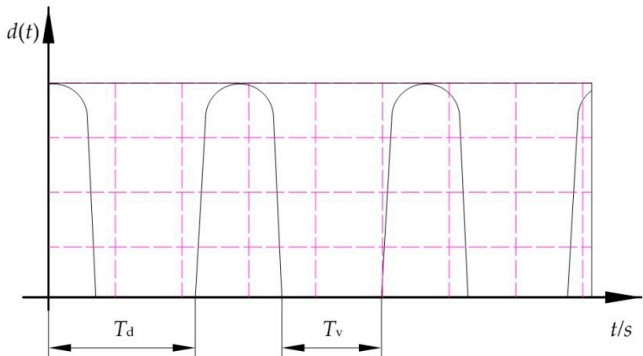

**Figure 3.** Fault rectangular pulse sequence produced by contact of the rolling elements with the spalling position.

In Figure 3, $d(t)$ is the maximum displacement of the bearing inner ring in the direction of bearing load, which can be measured experimentally, $T_d$ is the period of rectangular pulse sequence, and $T_v$ is the time the rolling element passes through the damage area. If the bearing outer ring is fixed and the inner ring rotating shaft rotates asynchronously, the position dimension of the damaged part is $L$; when the outer ring is damaged by spalling, $T_d$ and $T_v$ are expressed as

$$T_v = \frac{60L}{\pi n\left(1 - \frac{d}{D}\cos\alpha\right)} \tag{11}$$

$$T_d = \frac{1}{f_{out}} \tag{12}$$

where $n$ is the bearing speed.

As the bearing rotates at a high speed, the position of the fault point varies periodically with the bearing motion, so the impulsive force generated when the rolling element passes through the

load area is modulated by the load distribution. In a case where the pulse interval is very short, the modulated impulse $f_{\text{fault}}$ can be expressed as [32]:

$$f_{\text{fault}} = d(t) \cdot q(t). \tag{13}$$

*2.4. Nonlinear Dynamic Model*

Considering the radial clearance, time-varying stiffness, load distribution, fault pulse sequence, and other factors discussed above, the damping has little effect on the dynamic characteristics of the bearings and is regarded as a constant. The expression of the vibration analysis model of the fault ball bearing is

$$m\ddot{x} + C\dot{x} + Kx = F + f_{\text{fault}} \tag{14}$$

where $m$ is the quality of ball bearings, $F$ is the external load, and $C$ is the damping coefficient of ceramic materials; $C$ obtained from document [36].

Combining formula Equations (3)–(5), (9) and (11)–(14), a nonlinear dynamic model of full ceramic bearings with spalling fault on the outer ring can be obtained:

$$m\ddot{x} + C\dot{x} + Kx = F + d(t)Q_{\text{max}}\left[1 - \frac{2\delta_{\text{max}}}{2\delta_{\text{max}} - c}(1 - \cos 2\pi f_a t)\right]^{\frac{3}{2}}. \tag{15}$$

## 3. Time-Varying Stiffness of Outer Ring Crack Location

Assuming that the outer ring of full ceramic bearings contains cracks, as shown in Figure 4, the main causes of cracks are bending load and tensile load. Considering the effect of bearing seat on bearing, ignoring the effect of tensile stress on outer ring, the paper assumes that the main cause of cracks is bending load; then, the applied bending moment will not only promote crack propagation, but also increase the strain energy of structure.

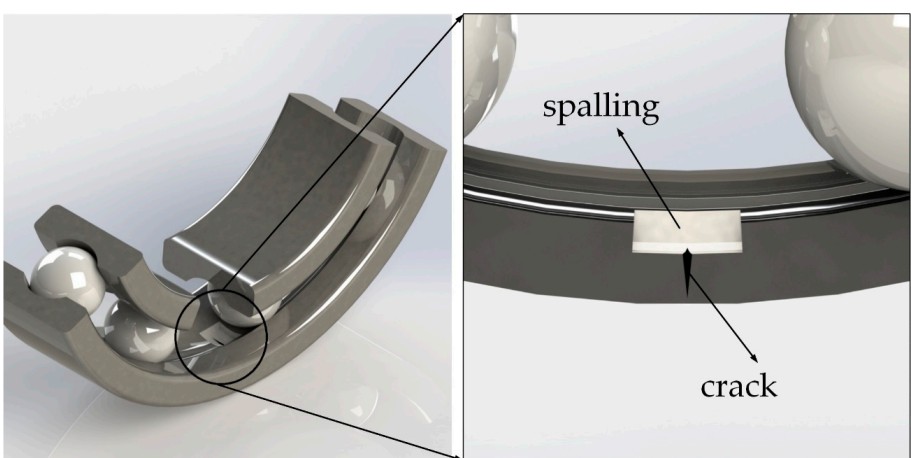

**Figure 4.** A schematic cross-sectional view of a crack at a spalling fault location in the ceramic bearing outer ring.

The final strain energy generated on the outer ring can be decomposed into the sum of the strain energy of the outer ring and the strain energy generated by the crack propagation in the crack-free state because the external action moment caused by the crack propagation does not change [33]:

$$W = EN_C + \Delta U = 2\Delta U \tag{16}$$

$$EN_C = \Delta U \tag{17}$$

where $EN_C$ is the energy for crack growth and $\Delta U$ is the increase of elastic strain energy in a cracked outer ring. The final strain energy of the cracked outer ring is

$$U_C = U + \Delta U = U + EN_C \qquad (18)$$

where $U$ is the strain energy of bearing outer ring in the absence of cracks.

For the uncracked outer ring with bending stiffness $EI$, subjected to a pure bending moment $M$, the strain energy in the outer ring is given by

$$U = \frac{1}{2} \int \frac{M^2}{EI} dx. \qquad (19)$$

According to the load distribution, the bending moment of the crack at the spalling position can be calculated as follows:

$$M = \int Q_\varphi \cdot r \cos \varphi d\varphi. \qquad (20)$$

The energy consumed for crack growth, based on fracture mechanics considerations, is given by

$$EN_C = \int (SERR) dA \qquad (21)$$

where $SERR$ is the strain energy release rate. Under the bending moment, the strain energy release rate $SERR$ could be expressed as

$$SERR = \frac{K_I^2}{E} \qquad (22)$$

where $E$ is the modulus of elasticity of the ceramic material; $K_I$ is that gravitational intensity factor of the crack calculated by the crack opening displacement method, the crack developing from the spalling is a mode I open crack, the cross-sectional view of the crack is shown in Figure 5 and according to the fracture mechanics of the ceramic material, the calculation of expression is as follows [30]:

$$K_I = M_e S_b \sqrt{\frac{\pi a}{Q}} \qquad (23)$$

where $M_e$ is the surface correction factor, which is difficult to calculate: $S_b = 3M/bt^2$.

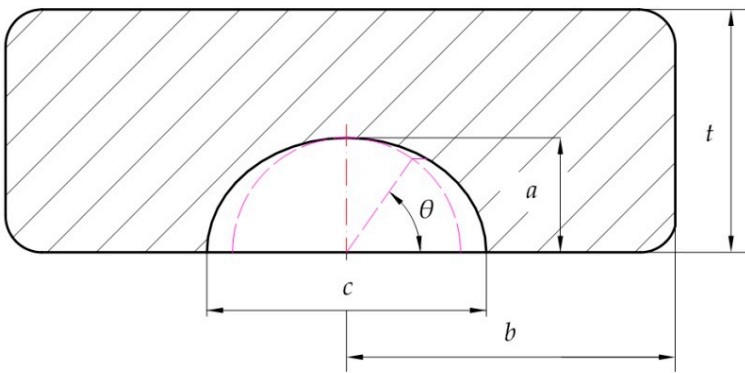

**Figure 5.** A schematic local cross-sectional view of the location of cracks in the outer ring.

In general, the surface correction factor $M_e$ is related to the ellipticity $a/c$ of the crack, the crack–outer ring thickness ratio $a/t$, the crack length–outer ring width ratio $c/b$, and the crack front position $\theta$. Newman and Raju give the expression of the surface correction factor [30]:

$$M_e = H\left(\frac{a}{c}, \frac{a}{t}, \phi\right) \ F\left(\frac{a}{c}, \frac{a}{t}, \frac{c}{b}, \phi\right). \qquad (24)$$

The function $H(\ )$ and $F(\ )$ in the formula can be calculated by the reference. In addition, it should be noted that *a/t, a/c, and c/b* have corresponding value ranges. $Q$ is the crack shape factor. According to the literature, we can get:

$$Q = 1 + 1.464\left(\frac{a}{c}\right)^{1.65}. \tag{25}$$

Simultaneous Equations (16)–(25) available:

$$\frac{1}{2}\int \frac{M^2}{EI_C}dx = \frac{1}{2}\int \frac{M^2}{EI}dx + \int (SERR)dA \tag{26}$$

where $EI_C$ is the flexural stiffness of full ceramic bearing outer rings in the event of cracks.

From the above equation, the modified bending stiffness of the cracked outer ring is obtained as

$$EI_C = \mu EI, \tag{27}$$

where $\mu$ is the crack damage coefficient.

According to the structural dynamics, the bending moment $EI$ is equivalent to the contact stiffness of outer ring:

$$K = f(EI). \tag{28}$$

Therefore:

$$K_{oc} = \mu K_{o1}. \tag{29}$$

In this paper, the crack–outer ring thickness ratio a/t and the crack length–outer ring width ratio c/b are taken as variables to study the effect of different scale cracks on the crack damage coefficient $\mu$. The effect of the crack depth a on the crack damage coefficient $\mu$ is studied when the crack length–outer ring width ratio c/b is 0.4 and the crack–outer ring thickness ratio a/t is 0.4, 0.5, 0.6, 0.7, and 0.8, respectively. The effect of crack width c on the crack damage coefficient $\mu$ is studied when the crack–outer ring thickness ratio a/t is 0.4 and the crack length–outer ring width ratio c/b is 0.4, 0.5, 0.6, 0.7, and 0.8, respectively. The effect of the change in crack depth the crack–outer ring thickness ratio a/t and the crack length–outer ring width ratio c/b on the crack damage coefficient $\mu$ of outer ring stiffness is shown in Figure 6.

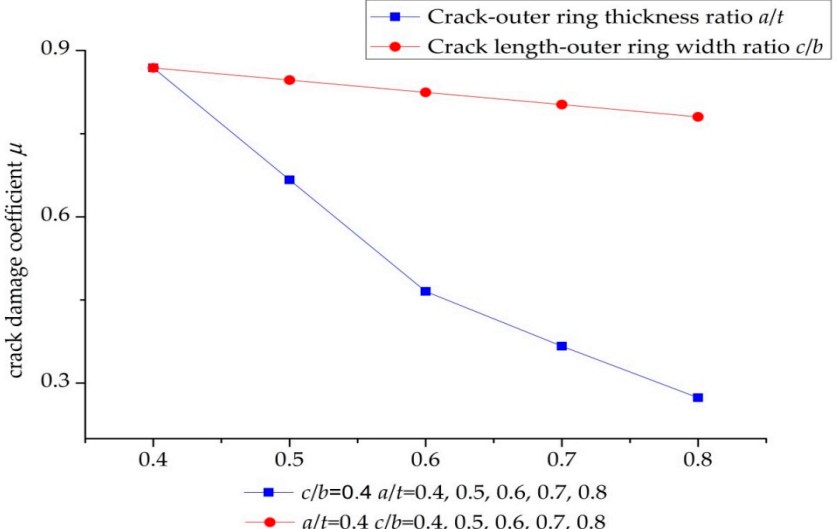

**Figure 6.** Comparison of the effects of another influencing factor on the crack damage coefficient $\mu$ when the crack–outer ring thickness ratio *a/t* or the crack length–outer ring width ratio *c/b* does not change.

From Figure 6, we can see that the influence of the crack–outer ring thickness ratio $a/t$ on the crack damage coefficient $\mu$ is much greater than that of the crack length–outer ring width ratio $c/b$ on the crack damage coefficient $\mu$, and the crack–outer ring thickness ratio $a/t$ weakens the crack damage coefficient $\mu$ greatly in the range of 0.4–0.8, while the crack length–outer ring width ratio $c/b$ weakens the crack damage coefficient $\mu$ stably relative to the crack–outer ring thickness ratio $a/t$.

Ceramic materials have high brittleness and low plasticity, so when calculating the cracks in the spalling damage of full ceramic bearing outer ring, it is assumed that the load distribution of outer ring in the state of spalling loss is consistent with that in the state of spalling crack. According to Hertz contact theorem and the load distribution in the case of spalling damage in the outer ring, the relative deformation of the outer ring in the process of spalling-induced crack can be calculated when only the spalling damage exists:

$$\delta_{c} = \left( \frac{Q_j}{K_{oc}} \right)^{-3/2}. \tag{30}$$

In the establishment of the equivalent dynamics of the outer ring of bearings, the deformation of the outer ring can be calculated as the superposition of the deformation of the outer ring with only the spalling damage and the relative deformation of the outer ring after the spalling damage develops into cracks:

$$\delta_{oT} = \delta_{os}\delta_{c}. \tag{31}$$

According to the Hertz contact theorem, the equivalent stiffness of the cracks in the spalling damage state of the outer ring can be obtained by:

$$K_{oT} = \frac{Q_\varphi}{\delta_{oT}{}^{3/2}}. \tag{32}$$

The contact stiffness of the full ceramic bearing can be obtained by bringing Equation (32) into Equation (9):

$$K_T = \left[ \frac{1}{(1/K_i)^{2/3} + (1/K_{oT})^{2/3}} \right]^{3/2}. \tag{33}$$

By introducing the calculated equivalent stiffness into the nonlinear dynamic model, the nonlinear dynamic model of full ceramic bearing outer ring with cracks in spalling can be obtained by:

$$m\ddot{x} + C\dot{x} + K_Tx = F + d(t)Q_{max}\left[1 - \frac{2\delta_{max}}{2\delta_{max} - c}(1 - \cos 2\pi f_a t)\right]^{\frac{3}{2}}. \tag{34}$$

## 4. Model Validation

### 4.1. Analog Simulation

Full ceramic bearing 7009AC is simulated under the condition that the bearing mainly bears radial load $F = 100$ N, the spalling size of bearing outer ring is set to $L = 0.$ 4 mm, the crack–outer ring thickness ratio $a/t$ is 0.4, 0.5, 0.6, 0.7, and 0.8, respectively, and the crack length–outer ring width ratio $c/b$ is 0.4, 0.5, 0.6, 0.7, and 0.8, respectively. The fourth-order Runge–Kutta algorithm of MATLAB is used to solve Equations (15) and (34). Then, the Fourier transform is performed on the simulated fault signal and the frequency domain of acceleration vibration spectrum of the model is obtained. The failure signals of different relative depth cracks and the cracks under different conditions in the spalling failure of the outer ring of full ceramic bearings are analyzed and compared with those of the cracks only in the spalling failure of the outer ring. The actual measure dynamic characteristics of that bearing in the spindle are compared with the simulation data of the crack fault bearing, and the crack generated at the position where the bearing spalling occurs can be effectively detected. The flow chart of monitoring cracks in the spalling position of the outer ring of the full ceramic bearing is shown

in Figure 7. The photo of the measurement and applications are shown in Figure 8a–c, respectively. The schematic diagram of the measurement and applications are shown in Figure 8d. As shown in Figure 8d, the ball bearing measured by the sensor are located in the spindle.

The crack size variables used in the simulation of faulty bearings are the crack–outer ring thickness ratio *a/t* and the crack length–outer ring width ratio *c/b*; these two variables are important parameters to measure and calculate the crack size [26]. At the same time, the ratio of the crack depth *a* and the crack width *c* to the thickness *t* and the width *b* of the bearing outer ring can reflect the damage degree of the crack to the bearing outer ring more clearly.

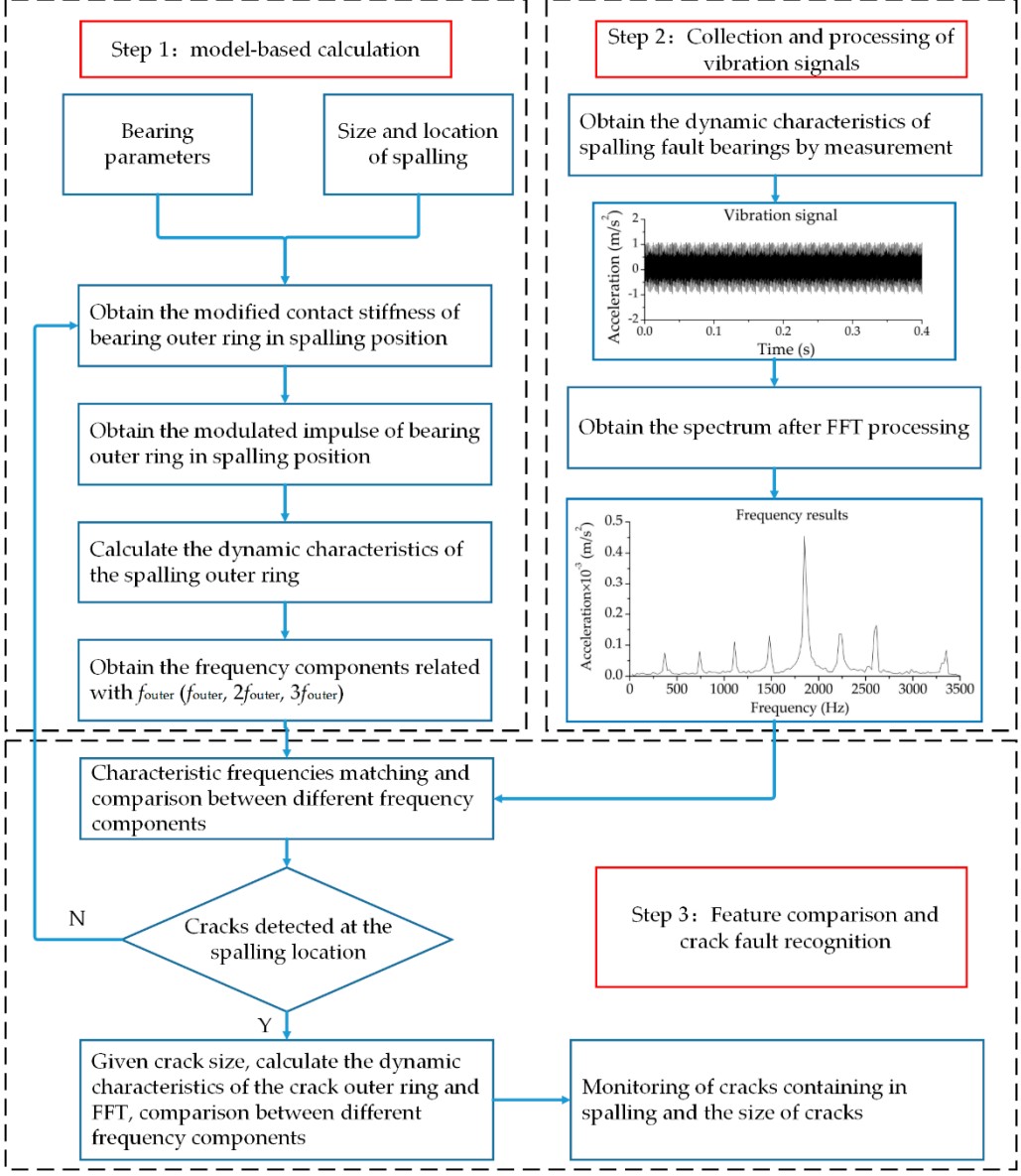

**Figure 7.** Flow chart of spalling position crack monitoring for full ceramic bearing outer ring.

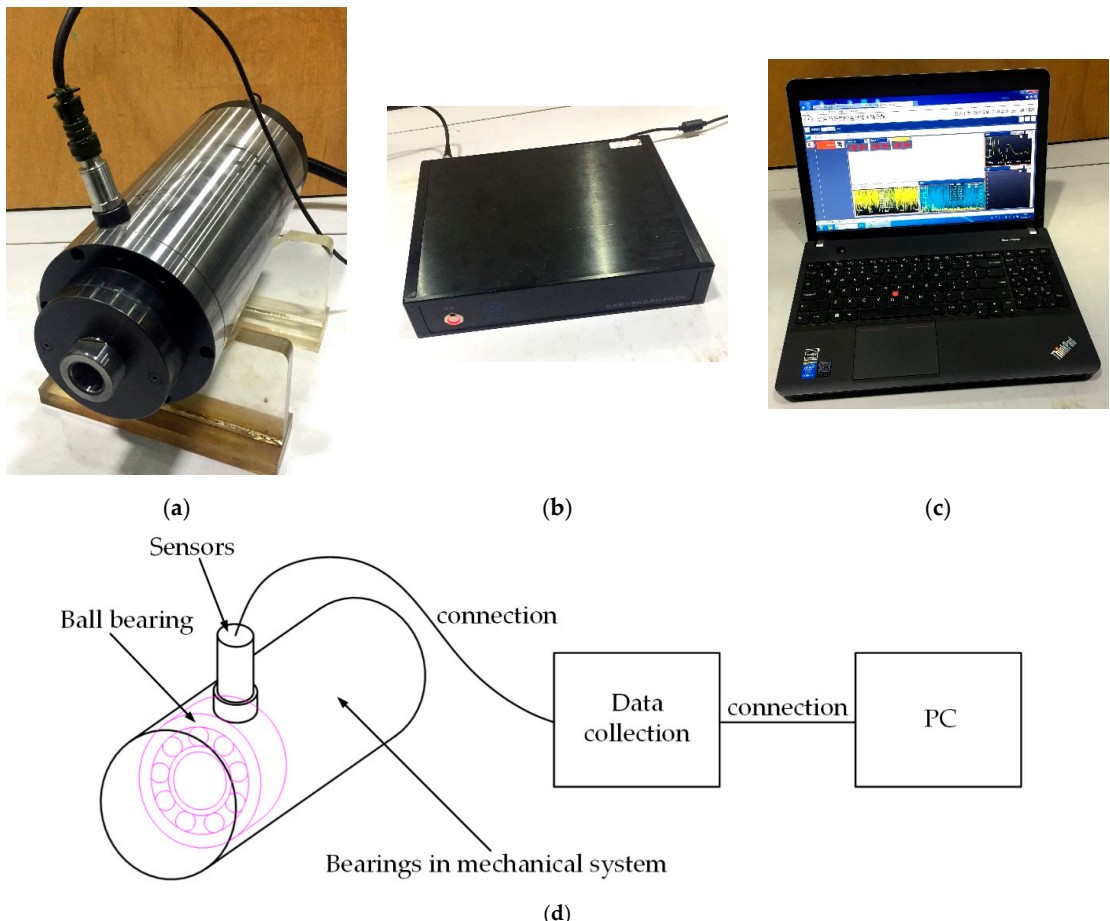

**Figure 8.** The photo and schematic diagram of the measurement and applications: (**a**) Photo of sensors and bearings in mechanical system; (**b**) photo of data collection; (**c**) photo of PC; (**d**) schematic diagram of measurements.

### 4.2. Effects of Different Relative Depths and Width

Figure 9 shows a comparison of time domain signals of the bearing vibration response in the case where the outer ring of the full ceramic bearing contains only the spalling fault and in the case where the crack depth $a$ and the crack width $c$ in spalling are the main influencing factors of the crack size, respectively. When the crack depth potential is the main factor, the crack–outer ring thickness ratio $a/t$ is 0.8 and the crack length–outer ring width ratio $c/b$ is 0.4; when the crack width is the main factor, the crack–outer ring thickness ratio $a/t$ is 0.4 and the crack length–outer ring width ratio $c/b$ is 0.8.

Comparing Figure 9a with Figure 9b,c, respectively, it can be seen that the simulated fault signal shows obvious periodic impact phenomena in the time domain during the crack propagation at the spalling fault, and that the acceleration signal of the vibration response increases significantly as it passes through the crack. Compared with Figure 9b,c, the acceleration signal of bearing vibration response increases more when the crack depth is the main influence factor, which indicates that the crack depth has a more significant influence on the movement of the outer ring of full ceramic bearing.

Because the shock phenomenon of the fault signal can only be seen roughly in the time domain, the vibration signal can be analyzed in more detail in the frequency domain. Figure 10a shows the comparison of the frequency domain signals of the bearing vibration response when the outer ring of the full ceramic bearing contains only the spalling fault and when the depth of the crack in the spalling is the main factor influencing the crack size. Figure 10b shows the comparison of the frequency domain signal of the bearing vibration response in the case where the crack width in spalling is the

main influence factor of the crack scale with the frequency domain signal of the vibration response in the case where only the spalling fault is included.

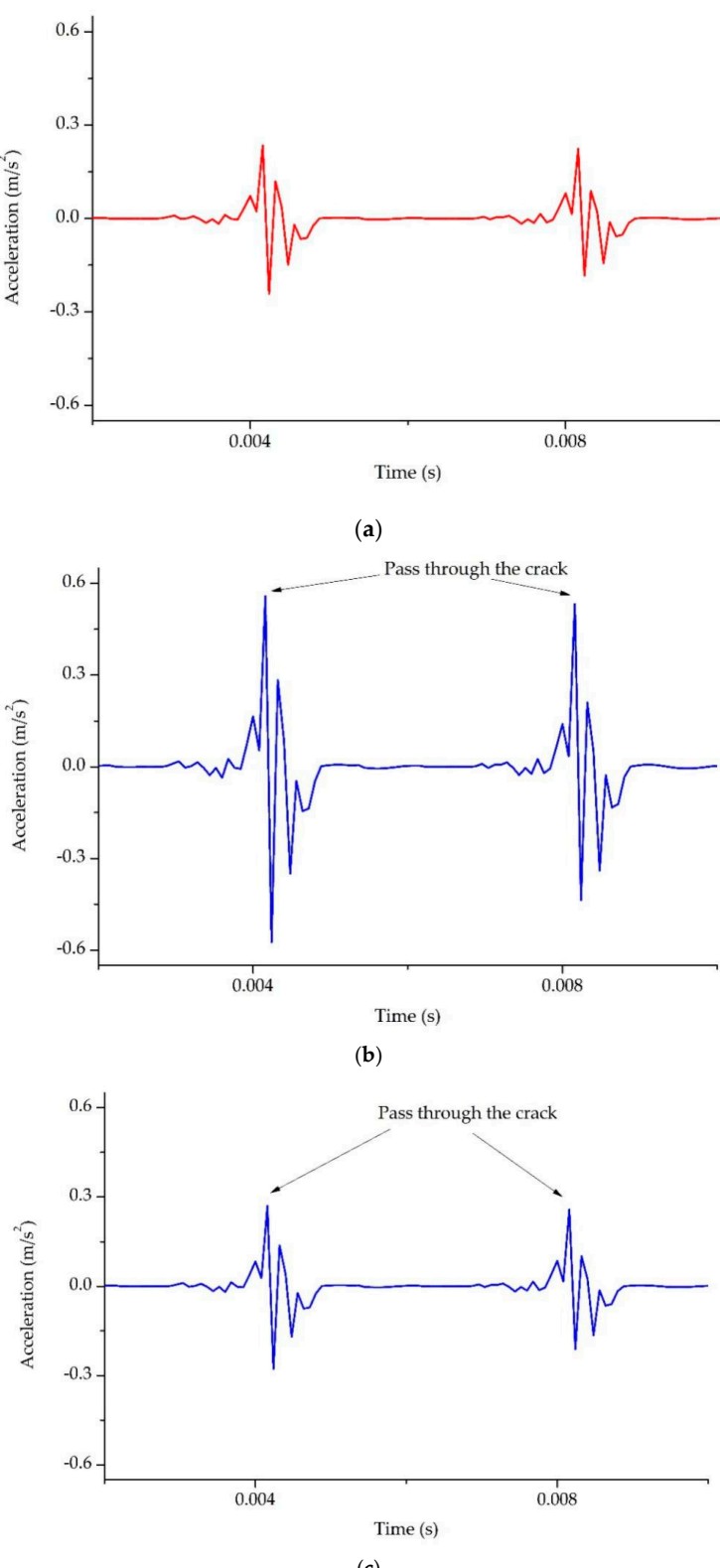

**Figure 9.** The time domain signals of bearing vibration response with (**a**) spalling faults only ($L = 0.4$ mm); (**b**) crack depth potential as the main influencing factor ($a/t = 0.8$ $c/b = 0.4$); (**c**) crack width potential as the main influencing factor ($a/t = 0.4$ $c/b = 0.8$).

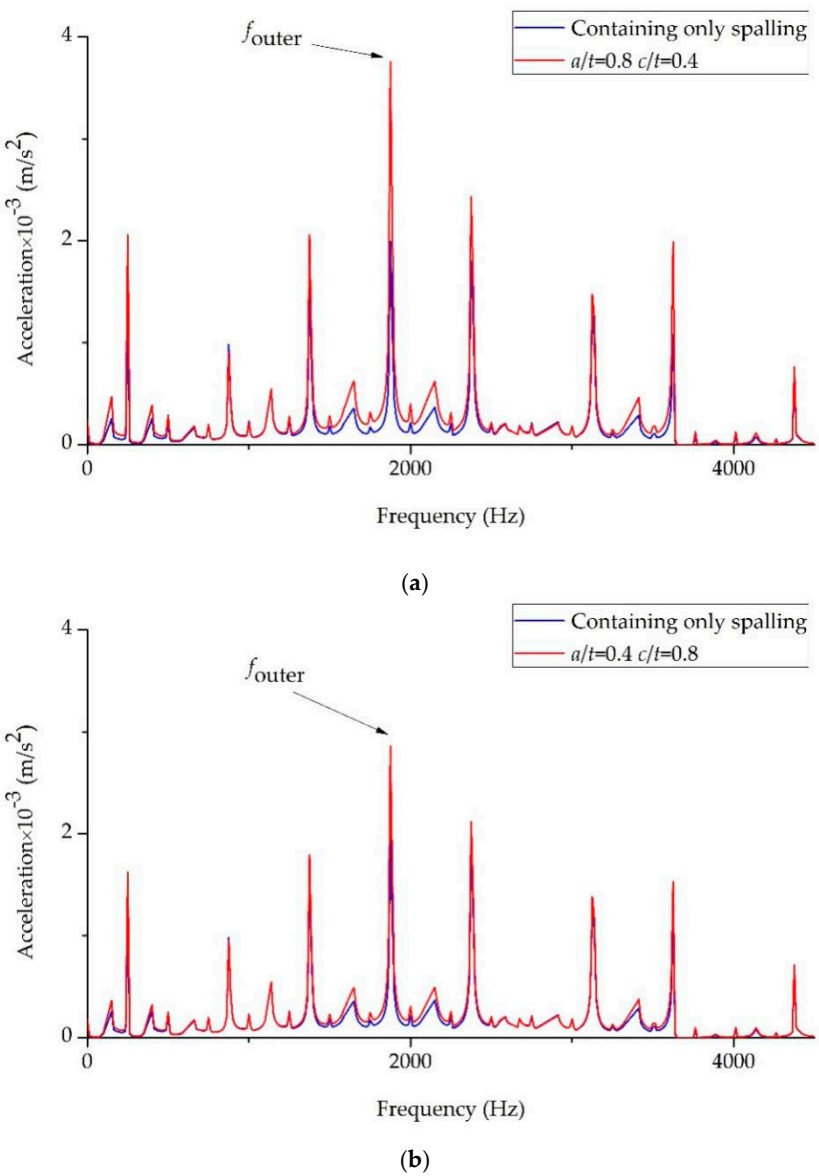

(a)

(b)

**Figure 10.** The frequency domain signals of bearing vibration response with (**a**) spalling faults only and crack depth potential as the main influencing factor (*a/t* = 0.8 *c/b* = 0.4); (**b**) spalling faults only and crack width potential as the main influencing factor (*a/t* = 0.4 *c/b* = 0.8).

As can be seen from Figure 10a,b, in the case where a crack is generated at the spalling point compared to the case where only the spalling fault is present, the amplitude of the fault characteristic frequency of the analog signal increases obviously. It shows that the phenomenon of cracking during spalling does aggravate the degree of bearing failure. Comparing Figure 10a with Figure 10b, it can be found that the increase of fault characteristic frequency is much larger when the crack depth accounts for the main influencing factor of the crack scale than when the crack width accounts for the main factor of the fault characteristic frequency. The amplitude of the fault characteristic frequency of the bearing outer ring increases with the increase of the crack depth and width, so the fault types contained in the outer ring can be judged more comprehensively by the amplitude corresponding to the fault characteristic frequency of the outer ring on the vibration response frequency domain signal of the bearing operation.

### 4.3. Influence of Crack Depth and Width

Figure 11 is the time-domain and frequency-domain plot of the vibration response of a full ceramic bear moving at the crack length–outer ring width ratio $c/b = 0.4$ with the crack–outer ring thickness ratio $a/t$ of 0.4, 0.5, and 0.6, respectively. Figure 12 is the time-domain and frequency-domain plot of that vibration response of ball bearing moving at the crack length–outer ring width ratio $c/b = 0.6$ with the crack–outer ring thickness ratio $a/t$ of 0.4, 0.5, and 0.6, respectively. Figure 13 is the time-domain and frequency-domain plot of that vibration response of ball bearing moving at the crack length–outer ring width ratio $c/b = 0.8$ with the crack–outer ring thickness ratio $a/t$ of 0.4, 0.5, and 0.6, respectively.

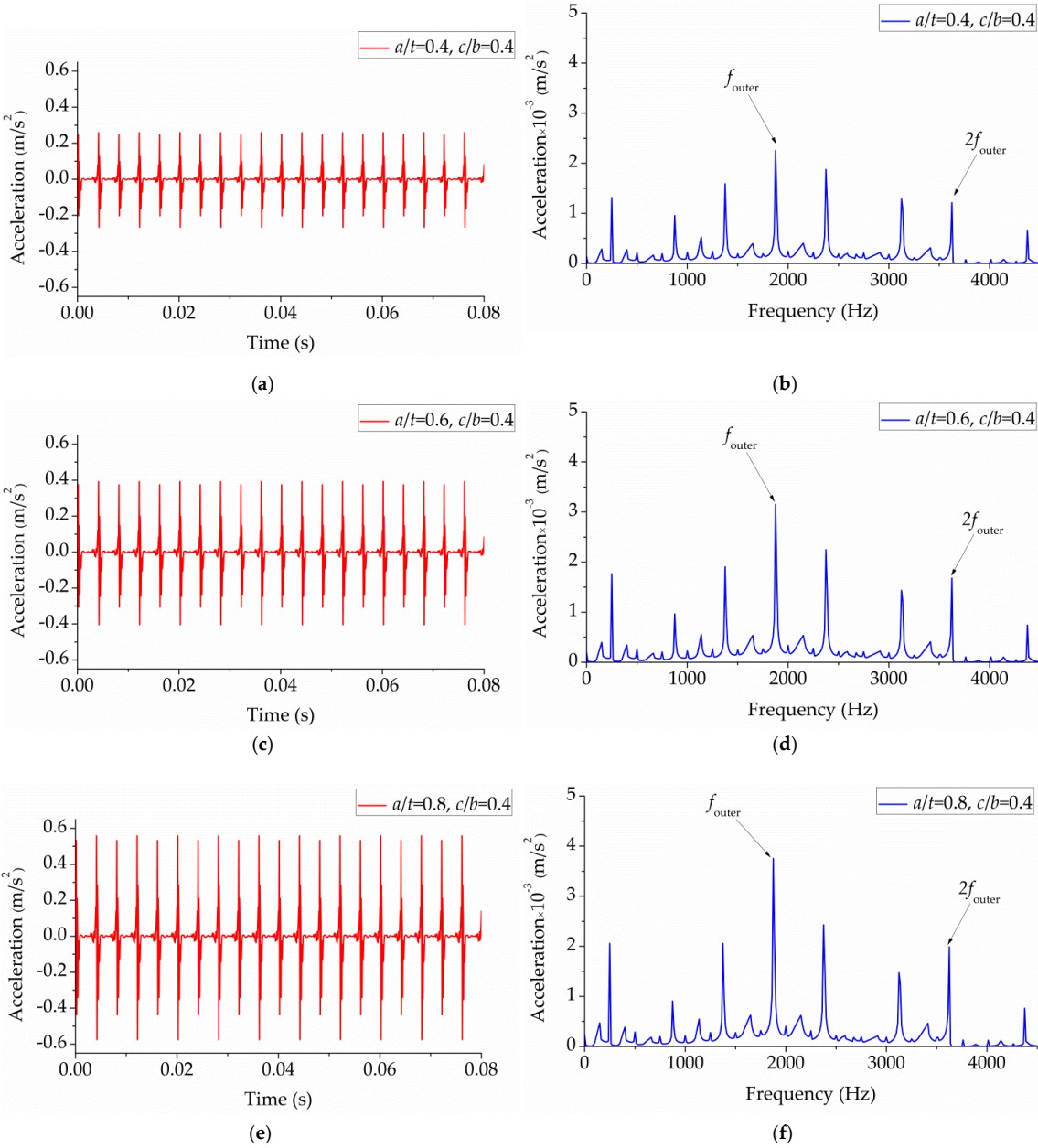

**Figure 11.** $c/b = 0.4$ time domain and frequency domain curves of vibration signals corresponding to different $a/t$: (**a,b**) Time-domain and frequency-domain response signals in case $a/t = 0.4$ $c/b = 0.4$; (**c,d**) time-domain and frequency-domain response signals in case $a/t = 0.6$ $c/b = 0.4$; (**e,f**) time-domain and frequency-domain response signals in case $a/t = 0.8$ $c/b = 0.4$.

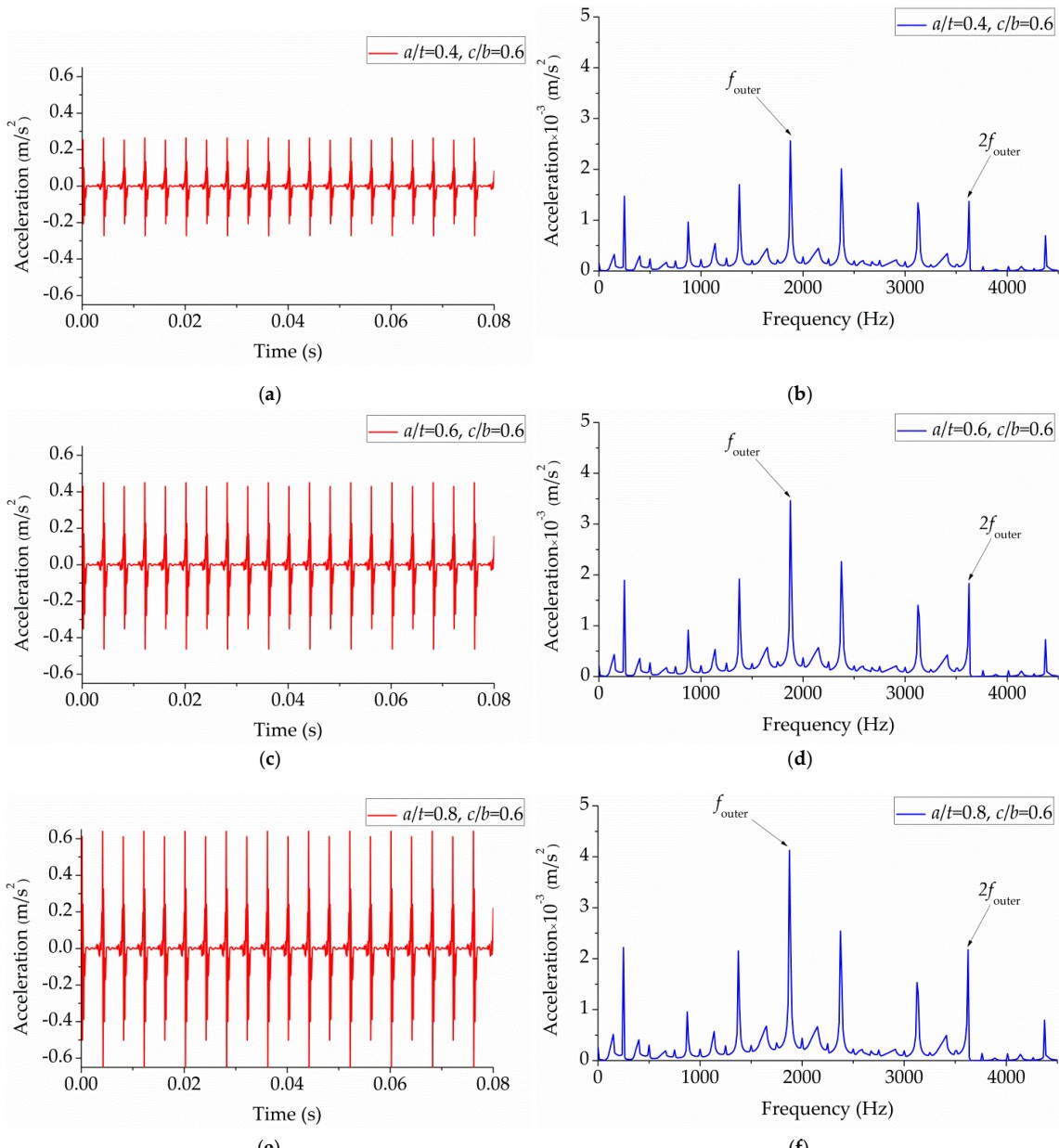

**Figure 12.** *c/b* = 0.6 time domain and frequency domain curves of vibration signals corresponding to different *a/t*: (**a**,**b**) Time-domain and frequency-domain response signals in case *a/t* = 0.4 *c/b* = 0.6; (**c**,**d**) time-domain and frequency-domain response signals in case *a/t* = 0.6 *c/b* = 0.6; (**e**,**f**) time-domain and frequency-domain response signals in case *a/t* = 0.8 *c/b* = 0.6.

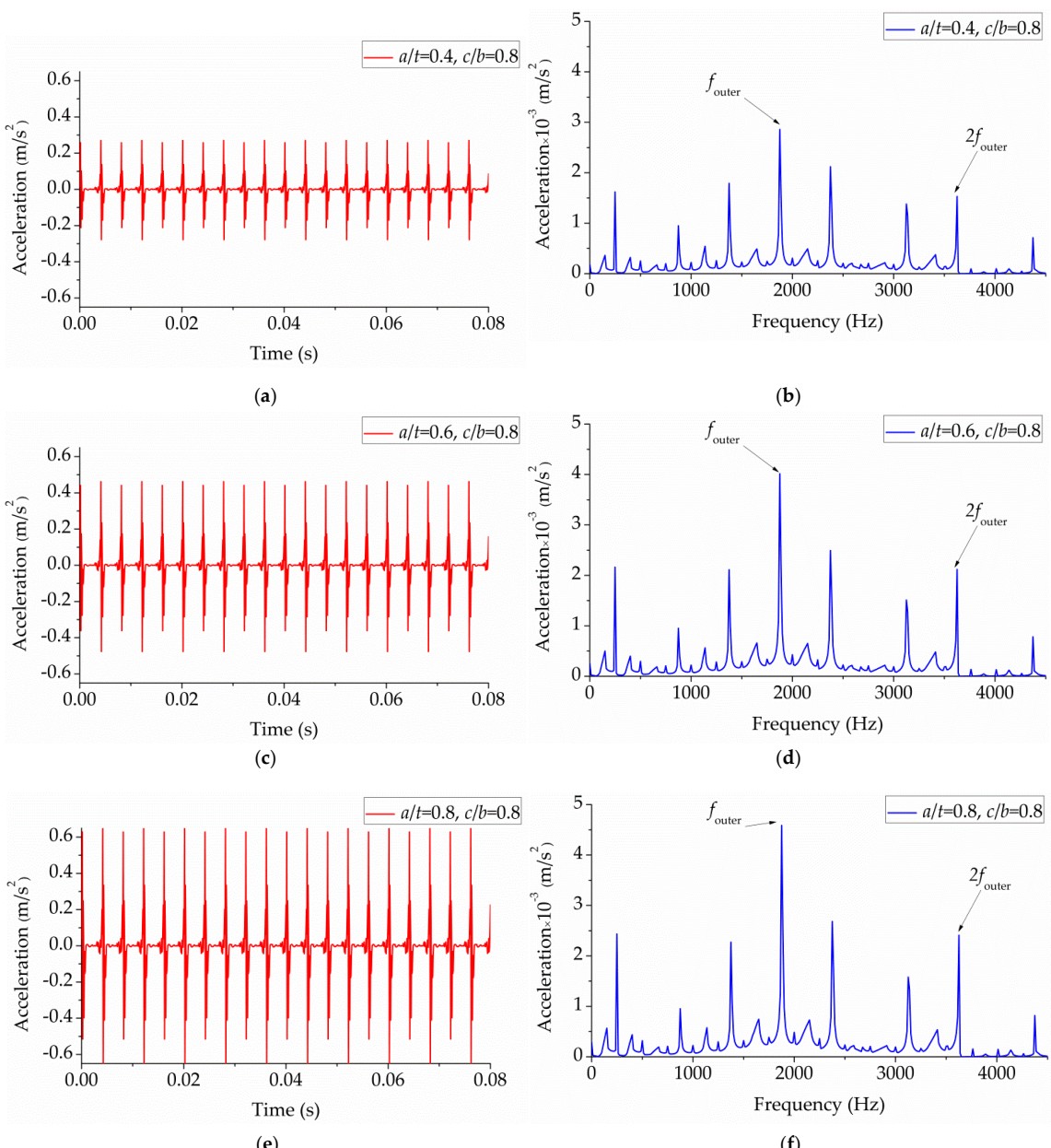

**Figure 13.** *c/b* = 0.8 time domain and frequency domain curves of vibration signals corresponding to different *a/t*: (**a**,**b**) Time-domain and frequency-domain response signals in case *a/t* = 0.4 *c/b* = 0.8; (**c**,**d**) time-domain and frequency-domain response signals in case *a/t* = 0.6 *c/b* = 0.8; (**e**,**f**) time-domain and frequency-domain response signals in case *a/t* = 0.8 *c/b*=0.8.

As can be seen from Figures 11–13, the vibration peak value corresponding to the crack passing through the time domain curve of the vibration signal increases significantly with the increasing crack depth. Although the vibration peak value also increases to a certain extent with the increasing crack width, there is no obvious impact phenomenon caused by the change of the depth. As that impact phenomenon of the fault signal cause by the crack generation can only be seen roughly in the time domain and the change of the vibration signal can be analyzed in detail in the frequency domain, only the expression of the vibration signal in the frequency domain is taken into account in the study of the effect of the relative depth of the crack on the vibration response.

Figure 14a shows the amplitude corresponding to the fault characteristic frequency of the analog signal under the different the crack–outer ring thickness ratio *a/t* conditions when the crack length–outer

ring width ratio *c/b* is constant. Figure 14b shows the amplitude corresponding to the fault characteristic frequency of the analog signal under different the crack length–outer ring width ratio *c/b* conditions when the crack–outer ring thickness ratio *a/t* is a constant.

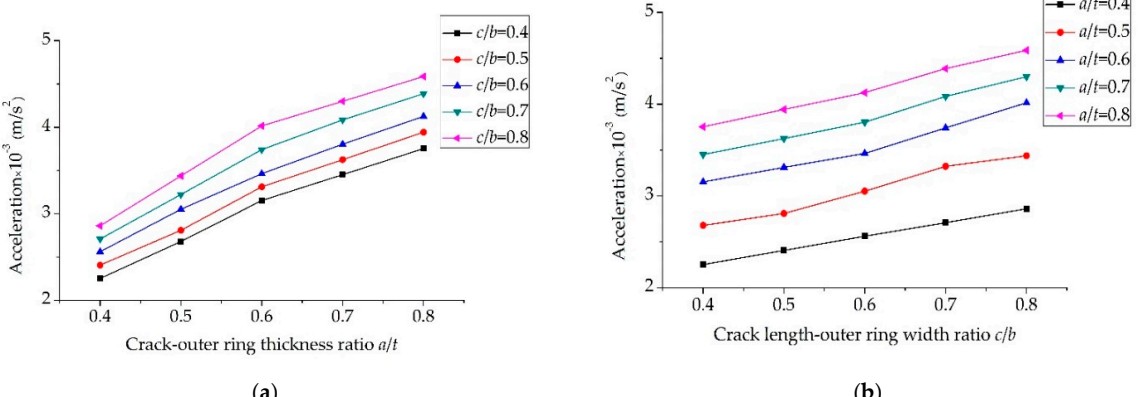

**Figure 14.** Fixing *a/t* or *c/b* and changing the amplitude comparison corresponding to the fault characteristic frequency obtained by another scale: (**a**) The amplitude corresponding to the fault characteristic frequency of the analog signal under different *a/t* conditions when *c/b* is constant; (**b**) the amplitude corresponding to the fault characteristic frequency of the analog signal under different c/b conditions when *a/t* is constant.

Comparing Figure 14a with Figure 14b, it can be seen that the amplitude change of the bearing fault characteristic frequency caused by the change of the crack–outer ring thickness ratio *a/t* is more obvious, and obviously, the influence of the crack–outer ring thickness ratio *a/t* on the vibration response of the bearing outer ring is more significant than that of the crack length–outer ring width ratio *c/b*. As can be seen from Figure 14a, when the crack length–outer ring width ratio *c/b* remains unchanged and the crack–outer ring thickness ratio *a/t* increases, the amplitude corresponding to the fault characteristic frequency of the bearing increases sharply, reaches about *a/t* = 0.6, then the slope decreases, and finally tends to be flat. As can be seen from Figure 14b, the fault characteristic frequency of the bearing rises steadily with the change of the crack length–outer ring width ratio *c/b*. Before the crack–outer ring thickness ratio *a/t* and the crack length–outer ring width ratio *c/b* reach 0.6, the influence of the crack–outer ring thickness ratio *a/t* on the amplitude of the fault characteristic frequency of the bearing (the slope of the change of the fault characteristic frequency) is more significant than that of the crack length–outer ring width ratio *c/b*. After reaching 0.6, the influence of the crack–outer ring thickness ratio *a/t* on the amplitude of the fault characteristic frequency of the bearing decreases, and the influence of the crack length–outer ring width ratio *c/b* on the amplitude of the fault characteristic frequency of the bearing is more significant.

Figure 15 shows the amplitude variation of the bearing outer ring fault characteristic frequency corresponding to the cracks at different scales the crack–outer ring thickness ratio *a/t* and the crack length–outer ring width ratio *c/b*.

According to Figure 15, each combination of different scales the crack–outer ring thickness ratio *a/t* and the crack length–outer ring width ratio *c/b* corresponds to the amplitude of one fault characteristic frequency, so that the amplitude of each fault characteristic frequency can determine the cracks at a certain scale.

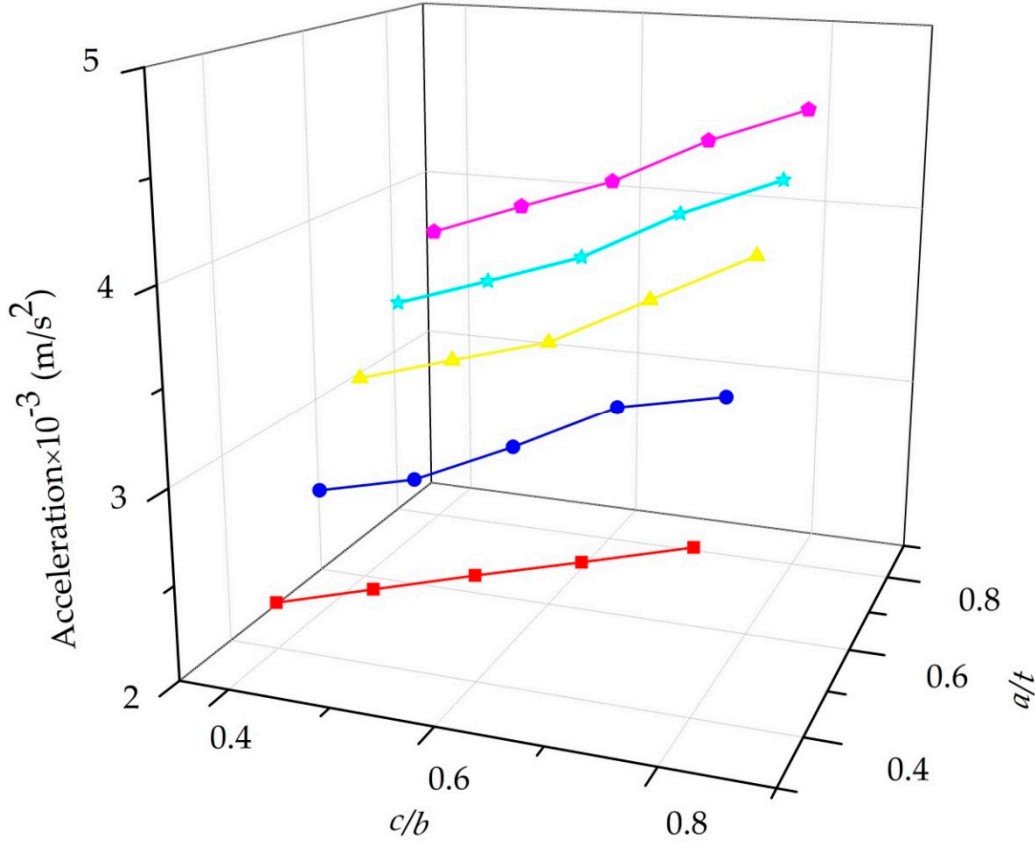

**Figure 15.** Amplitude curves of bearing outer ring fault characteristic frequencies corresponding to cracks at different *a/t* and *c/b* scales.

## 5. Discussion

Considering the influence of the modified rigidity of the outer ring and the pulse modulation of the contact load on the outer ring when the outer ring of full ceramic ball bearing spalling, a local spalling dynamic model of full ceramic ball bearing is established. The effect of spalling fault of the outer ring of the full ceramic ball bearing on dynamic vibration response of the system is studied.

According to the fracture mechanics theory, the change of the stress intensity factor at the crack site will lead to the increase of the crack strain energy and the increase of the final strain energy of the bearing outer ring, which will weaken the rigidity of the ceramic bearing outer ring. The rigidity of the outer ring decreases with the increase of the crack–outer ring thickness ratio and the crack length–outer ring width ratio. When the crack–outer ring thickness ratio and the crack length–outer ring width ratio increase by the same value, the rigidity of the outer ring decreases more with the increase of the crack–outer ring thickness ratio. However, when the values of the crack–outer ring thickness ratio and the crack length–outer ring width ratio increase beyond a certain threshold value, the weakening rate of outer rigidity of the crack–outer ring thickness ratio decreases significantly, while the weakening rate of outer rigidity of the crack length–outer ring width ratio remains stable.

The time-varying stiffness caused by crack initiation is introduced into the dynamic model; because the time-varying stiffness of the outer ring is reduced due to the generation of cracks, the amplitude of the acceleration signal of the vibration response of the outer ring increases significantly in the time-domain signal of the vibration of the outer ring. In the frequency domain signal of outer ring vibration, the fault characteristic frequency does not change obviously, but the peak value corresponding to the fault characteristic frequency increases significantly. Before the crack–outer ring thickness ratio and the crack length–outer ring width ratio reach a certain threshold value, the influence of the crack–outer ring thickness ratio on the amplitude of bearing fault characteristic frequency (slope



of change of fault characteristic frequency) is more significant than that of the crack length–outer ring width ratio. When the threshold value is reached, the influence of the crack–outer ring thickness ratio on the amplitude of bearing fault characteristic frequency decreases, while the influence of the crack length–outer ring width ratio on the amplitude of bearing fault characteristic frequency is more significant.

The characteristics of bearing vibration response waveform, which are obtained by simulation, can estimate the scale of cracking at the spalling fault to a certain extent and provide a theoretical basis for the health inspection of the full ceramic bearing outer ring during operation. In the future, the influence of temperature and stress field on the crack size will be considered. In addition, the damage of the outer ring of full ceramic bearings will also be studied from the aspect of crack propagation.

## 6. Conclusions

(1) With the appearance of the spalling fault on the outer ring, the vibration response signal of the faulty bearing will have more obvious periodic impact in the time domain, and the corresponding acceleration amplitude of the faulty feature of the outer ring will increase significantly in the frequency domain;

(2) Compared with the case where the outer ring has only spalling faults, the rigidity of the bearing outer ring will be weakened when there are cracks at the spalling position. Other influencing factors being equal, the rigidity of the bearing outer ring is weakened more obviously by the increase of the crack depth than by the increase of the crack width;

(3) The effect of crack depth on bearing vibration response is more significant when the crack–outer ring thickness ratio and the crack length–outer ring width ratio are less than 0.6, and the effect of crack depth on bearing vibration response is slower when the crack–outer ring thickness ratio and the crack length–outer ring width ratio are more than 0.6, while the effect of crack width on bearing vibration response is stable all the time;

(4) In the process of health inspection of full ceramic bearing outer ring, the scale of crack can be estimated by the amplitude of different fault characteristic frequency.

**Author Contributions:** Conceptualization, H.S. and X.B.; software, Z.L.; validation, Z.L., X.B., and Y.L.; formal analysis, H.S., X.B., and Y.W.; resources, H.S. and X.B.; data curation, Z.L.; writing—original draft preparation, Z.L.; writing—review and editing, H.S. and X.B.; supervision, H.S. and X.B.

**Funding:** This research was funded by the National Key R&D Program of China (2017YFC0703903), the National Natural Science Foundation of China (51705341, 51675353 and 51905357).

**Conflicts of Interest:** The authors declare no conflict to interest.

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
