# Peer review of "A Theoretical Model with the Effect of Cracks in the Local Spalling of Full Ceramic Ball Bearings"

_applsci, doi:10.3390/app9194142_

Round 1

Reviewer 1 Report

The authors report a detailed theoretical study on predicts the effect of cracks in the local spalling of full-ceramic rolling bearings. The authors explained the time-domain and frequency-domain vibration responses of bearings with flaking cracks on the outer rings of full-ceramic bearings under the condition of flaking.  This topic is of general interest and the paper clearly shows how computational techniques can give insights about the local damage and provide the basis for the fatigue damage and fault diagnosis of full-ceramic ball bearings.  However, several points need to be considered prior to acceptance.

Figs captions for Fig 1 to Fig 5 are not clear enough. Authors should explain it very details. Fig 11 is not clear enough. English of this paper need some improvement.

After the authors address the above mentioned points, the paper will be suitable for publication in Applied Science.

Author Response

请参阅附件。

Reviewer 2 Report

The paper is complete, very well balanced, supporter by evidencies.

Just few suggestions:

Row 72: in some application like in backup bearings the inner ring is fixed therefore It might be stated that outer ring usually is fixed.

Row 91: epsilon is often used to describe strain, perhaps another symbol could be' preferred.

Row 104 after raceway comma is followed by a capital letter...",Its"

Row 121 the literature is cited but no reference is expressively suggested as well as at row 193.

Very nice paper

Author Response

请参阅附件。
